# Within My Walls, I Escape Being Underestimated: A Systematic Review and Thematic Synthesis of Stigma and Help-Seeking in Dementia

**DOI:** 10.3390/bs15060774

**Published:** 2025-06-03

**Authors:** Marco Brigiano, Lara Calabrese, Ilaria Chirico, Sara Trolese, Martina Quartarone, Ludovica Forte, Alice Annini, Martino Belvederi Murri, Rabih Chattat

**Affiliations:** 1Department of Psychology, University of Bologna, 40126 Bologna, Italy; lara.calabrese3@unibo.it (L.C.); ilaria.chirico2@unibo.it (I.C.); sara.trolese@unibo.it (S.T.); martina.quartarone2@unibo.it (M.Q.); ludovica.forte@unibo.it (L.F.); alice.annini2@unibo.it (A.A.); rabih.chattat@unibo.it (R.C.); 2Department of Neuroscience and Rehabilitation, University of Ferrara, 44121 Ferrara, Italy; martino.belvederimurri@unife.it

**Keywords:** self-stigma, public stigma, help-seeking, dementia, stereotype, discrimination, shame, blame

## Abstract

Dementia-related stigma significantly influences help-seeking and affects the quality of care and support received by people with the condition. This review examines the impact of stigma on help-seeking among people with dementia and identifies key factors influencing this relationship. A systematic search across Web of Science, CINAHL, PsycINFO, PubMed, and Scopus identified seventeen qualitative studies that met the inclusion criteria. These criteria encompassed studies focusing on individuals aged 60 and older addressing public- or self-stigma and exploring help-seeking behaviors and related influencing factors. A thematic synthesis was employed to analyze the findings. The following five major themes emerged: reluctance to disclose the condition, internalization or rejection of stigmatizing beliefs, influence of family and community, attitudes of healthcare professionals, and lack of awareness in the broader society. Factors such as psychological decline, loss of autonomy, limited service access, peer support, and need for policy-level intervention were identified as central in shaping stigma. Findings related to the factors that influence this relationship indicate that stigma delays diagnosis and treatment, restricting access to adequate care. Both individual (e.g., autonomy, psychological well-being) and contextual (e.g., social networks, public policies) factors are crucial in moderating this dynamic. Targeted interventions addressing these dimensions are urgently needed to reduce stigma and facilitate timely help-seeking in dementia.

## 1. Introduction

Public awareness and understanding of dementia remain frequently limited or distorted, contributing to widespread stigmatization and representing a significant barrier to accessing appropriate care ([81]).

Stigma can adversely affect the lives of those living with dementia and their close ones in various ways, including the induction of shame, which can deter individuals from seeking medical assistance or participating in research ([59]; [69]; [52]). Stigma is acknowledged as one of the fundamental causes of health disparities influencing public policy and resource allocation, often resulting in the chronic underfunding of services dedicated to dementia care ([33]; [6]).

Stigma has been defined as a socially discrediting attribute or characteristic that encompasses negative beliefs, insufficient knowledge, and discriminatory behaviors, ultimately leading to unjust or unequal treatment ([5]; [30]). In the context of dementia, dementia-related stigma refers to the negative stereotypes, prejudices, and discriminatory behaviors directed toward people living with dementia, their caregivers, and their families ([9]; [44]; [34]). Stigma impacts an individuals’ ability to realize their potential and affects their experience and functioning according to their desires and talents throughout the illness ([23]). To better understand the impact of stigma on help-seeking of people with dementia, we need to analyze its relationship with social isolation, loneliness, and social interaction ([68]).

One of the most widely adopted frameworks in stigma research is the model proposed by [15] ([15]) and [17] ([17]). Within this framework, stigma is categorized into public stigma, referring to general societal attitudes and beliefs about individuals with the condition and their families, self-stigma, which involves the internalization of public attitudes and the associated negative consequences, and courtesy stigma, referring to the emotional responses and behaviors directed toward family members and professionals associated with the affected individual ([32]; [15]; [41]). This framework further conceptualizes stigma in terms of stereotypes, prejudice, and discrimination. Stereotypes involve generalized negative beliefs about a group, individuals with a specific condition, or oneself; prejudice is the emotional endorsement of these stereotypes, often expressed through negative emotional reactions toward others or oneself; and discrimination represents the behavioral response to prejudice, including exclusionary behaviors and diminished willingness to provide support ([78]; [51]; [82]). Although originally developed to understand stigma related to mental illness, this framework has been increasingly applied in studies examining stigma in the context of dementia ([57]; [28]; [77]).

Stigma plays a pivotal role in delaying diagnosis and therapeutic intervention as it often promotes denial, minimization, or ignorance of early symptoms ([27]; [74]). This may lead to delays in seeking help and reduced confidence ([63]; [31]; [49]; [66]), ultimately contributing to social isolation and loneliness ([5]; [10], [11]).

Help-seeking attitudes and behaviors are shaped by a complex interplay of available resources, personal beliefs, and cultural attitudes ([67]; [62]). This process involves several stages, including recognition of the problem; identification of the necessary resources; assessment of available resources; and the sharing of one’s condition and needs with others ([76]).

Two primary sources of help are typically identified, formal or professional help, which includes support from healthcare professionals or service providers, and informal help, referring to support from one’s social network, such as family, friends, or neighbors ([76]). Furthermore, help-seeking can be categorized by type: general (information, advice, support) or specific (diagnosis, treatment) ([60]; [76]).

Despite growing academic interest, significant gaps remain in the literature concerning stigma and dementia. The World Alzheimer Report ([4]) reiterated the widespread misconception of dementia as a normal aspect of aging, which continues to hinder early diagnosis and access to appropriate treatments.

A systematic review by [51] ([51]) highlighted the scarcity of studies that specifically examine the relationship between stigma and help-seeking. Among the 26 studies included, those focusing on self-stigma consistently reported delays in help-seeking behavior. Although not exclusively centered on this relationship, the review underscored stigma’s significant impact on the timing of help-seeking.

An earlier systematic review of 44 studies by [76] ([76]) had already identified stigma as a major barrier, emphasizing that certain beliefs (i.e., viewing dementia as a natural part of aging or considering care as the sole responsibility of the family) constitute real obstacles to seeking assistance.

To date, no study has systematically explored the perspectives of people with dementia regarding the link between stigma and help-seeking. This gap reflects broader societal neglect and highlights the urgent need for research that directly involves affected people. In this context, [55] ([55]) attempted to systematize the factors that hinder and facilitate the diagnostic process. Their findings indicate that stigma significantly impacts physical, psychological, and social well-being and can also act as a barrier to research participation, as evidenced by recruitment challenges. However, the review by [55] ([55]) does not focus solely on the experiences of people with dementia but also includes caregiver perspectives, who, according to the study, constituted the majority of participants. For this reason, our review aims to take a step further and consolidate existing evidence on the relationship between stigma and help-seeking in people with dementia, with an exclusive focus on the lived experiences of people with dementia.

The primary aim is to clarify the relationship between stigma and help-seeking, with particular attention paid to the factors that influence this connection. These may include individual characteristics (e.g., self-esteem, self-efficacy, cultural values) and social factors (e.g., support networks, community attitudes).

Specific objectives include the following:To analyze the impact of stigma on the help-seeking attitudes and behaviors of people with dementia.To identify the factors that influence the relationship between stigma and help-seeking behavior.

## 2. Materials and Methods

### 2.1. Search Strategy

The search strategy was developed using the PICO framework, a widely adopted tool in medical and systematic review research that facilitates the formulation of structured and well-defined research questions, thereby aiding the identification and evaluation of relevant evidence ([36]). In the present review, the PICO framework was applied to define the scope of the investigation, focusing on the following: people with dementia (population); their exposure to stigma (interest); no comparator was included (comparison); and the outcomes related to help-seeking attitudes, behaviors, and service utilization (outcome). The review protocol was prospectively registered on PROSPERO (CRD42024520847) and conducted in accordance with the PRISMA guidelines ([54]).

A systematic literature search was conducted across the following five major electronic databases: Web of Science, CINAHL, PsycINFO, PubMed, and Scopus. Search terms related to the population included ‘dementia’, ‘Alzheimer’s disease’, ‘neurocognitive disorder’, and ‘neurodegenerative disease’. The search strategy (see Table 1) was limited to peer-reviewed journal articles reporting empirical studies published in English up to 2 May 2025. No publication date restrictions were applied in order to maximize the breadth of the results.

The screening process was conducted in three stages, the removal of duplicates, the screening of titles and abstracts, and full-text review. All retrieved records were imported into Rayyan (v1.6.0) software ([53]), where duplicates were removed. In the second phase, two independent reviewers (M.B. and L.C.) screened titles and abstracts against pre-defined inclusion and exclusion criteria; discrepancies were resolved through discussion with a third reviewer (R.C.). In the final phase, full-text articles of potentially eligible studies were independently assessed by M.B. and L.C., with justifications for exclusion documented. Any disagreements were resolved through consensus with R.C. The final selection of studies was documented in an Excel spreadsheet, including key information such as authorship, year of publication, country, study design, sample size, and main findings. Following the screening process, no eligible quantitative studies were identified; therefore, only qualitative studies meeting the inclusion criteria were incorporated into the final synthesis.

### 2.2. Inclusion and Exclusion Criteria

To be eligible for inclusion in this systematic review, studies had to meet the following criteria: (1) focus on individuals aged 60 years or older with a diagnosis of dementia (i.e., the population of interest); (2) employ an empirical design (qualitative or quantitative); and (3) examine aspects related to stigma (public or self-stigma), help-seeking attitudes and behaviors among people with dementia, or the factors influencing the relationship between stigma and help-seeking. Studies were excluded if they (1) did not primarily focus on people with dementia; (2) were not original research articles (e.g., reviews, conference abstracts, opinion papers, or dissertations); or (3) were not available in full-text in English.

### 2.3. Quality Appraisal

The quality of the included studies was assessed using the critical appraisal skills programme ([19]) checklist for qualitative research (see Table 2). This tool includes ten items divided into the following three sections: (A) are the results valid; (B) what are the results; and (C) will the results help locally. The appraisal was conducted independently by two researchers (M.B. and L.C.), and any discrepancies were discussed and resolved. When necessary, a third researcher (R.C.) was consulted. Although quality assessment did not influence study inclusion, it was used to evaluate the robustness of the synthesized findings.

### 2.4. Data Extraction and Synthesis

Data were extracted independently by two researchers (M.B. and L.C.) using a standardized form capturing study objectives, methodology, sample characteristics, and findings related to stigma, help-seeking attitudes and behaviors, and relevant moderating factors. Disagreements were resolved by consensus in consultation with a third researcher (R.C.). No eligible quantitative studies emerged during the screening process; consequently, a thematic synthesis approach was adopted to analyze the qualitative data ([71]).

Thematic synthesis, as described by [71] ([71]), was employed to analyze the qualitative data. This method involves systematic coding of primary data and the generation of both descriptive and analytical themes, aiming to move beyond simple description toward higher-level conceptual interpretation while maintaining close linkage to the original data. To enhance transparency and methodological rigor, the synthesis was reported in accordance with the ENTREQ guidelines ([73]).

The synthesis process followed three stages: (1) line-by-line coding of study findings, where each line of text was inductively coded by assigning descriptive labels to units of meaning (e.g., “Understanding of dementia”, “Impact on the person living with dementia”); (2) development of descriptive themes, in which similar codes were grouped together to reflect patterns across studies (e.g., “Lack of awareness”, “Perceived stigma”); (3) generation of analytical themes, where interpretations extended beyond the primary data to address the review question and offer new conceptual insights (e.g., “Reluctance to Share One’s Condition and Risk of Labeling”, “From Public Perception to Internalization or Rejection of Stigmatizing Beliefs”). No software was used during the process.

To ensure the credibility and coherence of the thematic synthesis, regular team discussions were held to review and refine emerging themes.

Finally, we assessed confidence in each review finding using the GRADE-CERQual approach ([39]). This method evaluates the strength of qualitative evidence based on four criteria, methodological limitations of the contributing studies, relevance to the review question, coherence of the finding, and adequacy of the data (see Appendix A).

## 3. Results

### 3.1. Study Selection

A total of 5192 records were identified through database searches. All the studies were entered on Rayyan ([53]) and duplicated records were removed. After removing 708 duplicates, 4484 records remained for screening. During the screening of title and abstracts, 4450 records were excluded. Consequently, 34 full-text articles were assessed for eligibility, of which 17 studies were excluded (not outcome of interest, not population of interest, and full-text not available). A total of 17 studies met the inclusion criteria and were included in the qualitative synthesis. The entire selection process used to identify studies is presented as a PRISMA flow diagram (see Figure 1).

#### 3.1.1. Risk of Bias in Studies

Among the 17 studies included, 9 fully satisfied the Critical Appraisal Skills Programme (CASP) criteria, indicating high methodological rigor. The remaining studies displayed minor limitations, particularly regarding the clarity of recruitment strategies, the explicit consideration of researcher–participant relationships, the attention given to ethical considerations, and the clearness of statements of findings. Nonetheless, none of the studies were excluded on the basis of insufficient quality as each contributed substantively to the synthesis. An overview of all quality assessment domains is provided in Table 2.

#### 3.1.2. Study Characteristics

All 17 studies included in the synthesis employed qualitative methodologies and exclusively explored the perspectives of individuals living with dementia (see Table 3). Studies that did not explicitly indicate the involvement of people with dementia or where it was unclear whether their perspectives were solely represented were excluded. Collectively, these studies underscore the pervasive nature of dementia-related stigma and its substantial influence on help-seeking behaviors and pathways. During data extraction, it was noted that [7] ([7]) and [70] ([70]) adopted alternative conceptualizations of stigma, distinguishing between treatment stigma and identity stigma. These conceptualizations were consistent with the broader stigma literature and thus were deemed appropriate for inclusion.

The geographical distribution of the studies was as follows: Canada (one), New Zealand (one), United States (one), Australia (two), India (one), China (two), United Kingdom (four), Northern Ireland (one), Norway (one), Tanzania (one), Pakistan (one), and Israel (one). The qualitative methods employed included individual interviews and semi-structured interviews (n = 12), focus groups (n = 2), and a combination of interviews and focus groups (n = 3). Participant recruitment was conducted through memory services (n = 4), medical services (n = 5), the general public (n = 8), and one from a prior study (n = 1).

While the primary focus of this systematic review was on the lived experience of people with dementia, the studies included a range of participant types such as people with dementia (n = 195); dyads comprising people with dementia and their caregivers (n = 6); caregivers (n = 90); family members (n = 63); healthcare professionals (n = 57); community members (n = 26); people with young-onset dementia (n = 3); and caregivers of people with young-onset dementia (n = 4).

### 3.2. Thematic Synthesis

#### 3.2.1. Impact of Stigma on Help-Seeking

The following five key themes emerged from studies on the perspectives of people with dementia regarding the impact of stigma on help-seeking attitudes and behaviors: (1) reluctance to share one’s condition and risk of labeling; (2) from public perception to internalization or rejection of stigmatizing beliefs; (3) how family and community shape the experience of seeking-help of people with dementia; (4) professional attitudes and eligibility processes in dementia services; and (5) stigma stems from lack of awareness and knowledge of dementia.

##### Reluctance to Share One’s Condition and Risk of Labeling

One frequently reported barrier to accessing post-diagnostic services was the persistence of stigma and negative emotions associated with the condition. Individuals living with dementia expressed feelings of shame and insecurity linked to cognitive changes, which intensified their sense of self-doubt and fear of rejection, ultimately leading to anticipatory avoidance of social situations ([56]). Stigma has a detrimental impact on symptom recognition and deters people with dementia from disclosing their condition ([56]; [48]). Dementia is frequently described as a “hidden problem”, contributing to social isolation and limited access to care services ([7]; [72]). In Chinese cultural contexts, reluctance to disclose the condition, even to primary care physicians, highlights the broader stigmatic barriers impeding disclosure and help-seeking ([13]).

The perceived incurability of dementia further discourages help-seeking, generating psychological distress and fear surrounding the diagnosis ([40]). Concerns about stigmatization and negative labeling increase the risk of social exclusion and emotional withdrawal ([7]; [47]; [83]).

##### From Public Perception to Internalization or Rejection of Stigmatizing Beliefs

Several participants referred to the phenomenon of “perceived stigma”, a culturally widespread barrier that inhibits both the recognition of dementia and the pursuit of external support. This involves the awareness of societal beliefs and behaviors that marginalize those with dementia ([56]; [7]). People with dementia often internalize societal attitudes, manifesting as self-criticism, shame, and a devalued self-concept, marked by heightened sensitivity to others’ perceptions ([56]). This dynamic is conceptualized by [70] ([70]) as “identity stigma”.

Conversely, [47] ([47]) found that participants frequently sought to avoid being pitied or condescended to, what they referred to as receiving the “fool’s pardon”, suggesting that some individuals actively resist internalizing stigmatizing narratives.

##### How Family and Community Shape the Experience of Seeking-Help of People with Dementia

Many participants reported that stigmatizing attitudes were present within both familial and community contexts. These were often rooted in misconceptions about dementia or suspicions that individuals were exaggerating or fabricating symptoms, which delayed the initiation of help-seeking pathways ([37]; [83]). In one case a participant described being mocked by a community member ([79]), whilst others recounted experiences of stress induced by communal negativity ([40]). These experiences often led to denial of diagnosis and further isolation, exacerbated by relatives’ fears of social judgment ([37]).

However, some studies also reported supportive and empathetic family environments, with such attitudes often associated with higher educational attainment ([79]). These individuals, family members, and community members have the potential to support people with dementia in overcoming societal challenges or may inadvertently reinforce societal stigma surrounding dementia ([47]).

##### Professional Attitudes and Eligibility Processes in Dementia Services

Healthcare professionals were frequently perceived as lacking knowledge about available dementia care resources or expressing stigmatizing attitudes, which hindered access to appropriate services ([56]; [26]). Infantilizing behavior and a lack of empathy were recurrent concerns, with some individuals describing their condition as being viewed as a “death sentence” ([29]; [37]; [47]). In the Chinese context, [83] ([83]) described a professional attitude of nihilism, reflective of dementia’s common framing as a terminal condition. Such approaches foster disillusionment, damage trust in physician–family relationships, and deepen stigma.

Several participants highlighted systemic inequities in how dementia is treated relative to other conditions, noting that dementia is often given less consideration ([20]; [56]). Furthermore, eligibility assessments for long-term care institutions were frequently described as inadequate and offensive, resulting in feelings of alienation and indignation ([56]). These experiences contribute to what [70] ([70]) conceptualize as “treatment stigma”.

##### Stigma Stems from Lack of Awareness and Knowledge of Dementia

Help-seeking behaviors are significantly shaped by communal beliefs and attitudes toward dementia. Interactions with various societal actors influence the trajectory of care and the lived experience of the condition ([26]; [40]).

Public perceptions often reflect stereotypes that portray people with dementia as incompetent or tragic figures ([26]; [47]; [70]; [75]). These stigmatizing views are reinforced by media narratives, which frequently utilize dehumanizing language such as “sufferer”, “demented”, or “crazy” ([20]; [46]; [47]; [75]; [83]).

Participants emphasized that these portrayals intensify fear and reluctance to seek help, and they also noted the lack of systemic representation of people with dementia in care planning as an unmet need ([26]). Stigma was widely attributed to societal ignorance and insufficient public education about dementia ([20]). The conflation of dementia with general mental illness further exacerbates stigmatization ([37]), and in rural contexts, where awareness is even more limited, stigma may lead to severe outcomes, including neglect and abuse ([47]).

#### 3.2.2. Factors That Influence the Relationship Between Stigma and Help-Seeking Attitudes and Behaviors

In the context of the relationship between stigma and help-seeking behavior, the qualitative factors that affect the relationship refer to individual or contextual variables that influence the strength or direction of this relationship. These may include personal characteristics, such as the individual’s level of self-esteem, or cultural values, which may influence how stigma is perceived and internalized. In addition, social and environmental factors, such as the availability of support networks, cultural attitudes toward help-seeking, and the presence of stigma-related narratives within specific communities, can exacerbate or mitigate the impact of stigma on an individual’s willingness to seek help.

The following four key themes were identified from the studies: (1) the impact of psychological decline, isolation, and loss of autonomy on help-seeking; (2) a gap in accessible and supportive services; (3) caregiver support, professional relationships, and peer support; and (4) what government can do to tackle dementia.

##### The Impact of Psychological Decline, Isolation, and Loss of Autonomy on Help-Seeking Behaviors

As dementia progresses, individuals frequently express fear related to memory loss, reduced self-control, and increased dependence on familial caregivers ([48]; [46]). Many participants report a significant loss of independence following diagnosis, often feeling that their ability to perform even basic tasks autonomously is questioned by others ([47]). Participants frequently reported affective symptoms such as sadness, irritability, and anger, coupled with difficulties in accepting their altered cognitive status ([56]; [13]; [72]). Notably, some individuals expressed that they did not require social support ([56]).

Psychological deterioration, isolation, and a progressive loss of autonomy significantly impair the capacity of people with dementia to seek and access support. In [12] ([12]), psychological decline was reported to render previously familiar experiences unrecognizable. One participant, for instance, described how their inability to fulfill religious practices triggered a heightened awareness of cognitive deterioration ([12]). Such psychological responses often result in social withdrawal, profound loneliness, diminished self-confidence, and a compromised sense of self-worth, with repercussions extending beyond the individual to affect family systems ([20]; [37]; [48]; [75]).

Tangible implications of cognitive decline include revocation of their driving license, an experience commonly perceived as unjustified and emotionally distressing ([56]; [48]). As noted by [47] ([47]), maintaining independence in daily living remains a central concern for people with dementia, emphasizing the deep personal value attached to autonomy.

Importantly, not all consequences of diagnosis are negative. [48] ([48]) observed that receiving a diagnosis brought a sense of relief by offering a framework for understanding cognitive changes, thus facilitating help-seeking. Similarly, [20] ([20]) report an instance in which a participant established a private support group, illustrating the potential for agency and self-directed coping strategies post-diagnosis.

##### A Gap in Accessible and Supportive Services

A pervasive finding across studies is the lack of awareness among people with dementia regarding available local services, which complicates the navigation of healthcare systems ([20]; [56]). This informational gap contributes to delays between diagnosis and engagement with services ([7]; [29]; [37]). Barriers are particularly pronounced for non-native individuals, with ethnic minority status and the limited cultural competence of healthcare systems further impeding access ([20]; [56]; [7]).

These challenges are evident in primary care consultations, referrals to specialists (e.g., geriatricians), and access to dedicated dementia services, problems exacerbated by workforce shortages and insufficient dementia-specific training, especially in rural areas ([20]; [26]; [29]; [37]; [46]). A recurring concern raised by participants is the absence of a dedicated specialist or case manager to guide them through the post-diagnostic process ([20]; [56]).

Additionally, the limited availability of meaningful engagement opportunities and social activities was frequently reported ([29]). These factors contribute to delayed diagnoses, fragmented care, and inadequate service integration, underscoring a systemic failure to provide accessible and supportive care for people with dementia ([29]).

Physical barriers also restrict service utilization. Participants frequently cited inaccessible facilities that do not accommodate mobility limitations, and the financial burden associated with care ([37]). The shortage of qualified personnel and diagnostic resources further impairs screening and assessment. As [40] ([40]) noted, existing services often fall short of user expectations. In [75] ([75]), only one participant had undergone formal cognitive assessment, highlighting substantial service inadequacy. A sense of neglect from social services was frequently reported, especially in rural contexts ([83]).

This scarcity coincides with a prevalent preference for non-pharmacological and home-based care among people with dementia and their families, as opposed to institutional long-term care. This preference is informed by distrust, perceived insecurity, and dissatisfaction with available options ([20]; [70]; [75]; [83]). Furthermore, pharmacological treatments are often presented as the only option, with little to no information about alternative interventions or dedicated support services, highlighting a lack of comprehensive care planning ([56]).

##### Caregiver Support, Professional Relationships, and Peer Support Along Help-Seeking Path

A key facilitating factor for early diagnosis and subsequent help-seeking is the role of the caregiver. Caregivers often serve as the first to recognize early symptoms, initiate contact with health professionals, and navigate available screening services ([40]; [79]). Furthermore, [56] ([56]) highlight the importance of extended family networks, which provide emotional support, access to information, and assistance in organizing meaningful and enjoyable activities.

Positive relationships with healthcare professionals also facilitate help-seeking by mitigating the impact of stigma. People with dementia place great importance on being listened to and treated with dignity, which correlates with increased life satisfaction and more positive perceptions of care quality ([56]; [29]; [72]). Additionally, participants underscore the benefits of peer support, which further encourages engagement in the help-seeking process ([20]; [46]).

##### What Government Can Do to Tackle Dementia and to Promote Help-Seeking

Among the most influential factors facilitating help-seeking and mitigating stigma are those tied to governmental policy and public health initiatives. A central recommendation across studies involves the implementation of targeted public awareness campaigns that educate the population about dementia, emphasize the benefits of timely diagnosis, and clarify the pathways to access appropriate services ([40]; [75]). These campaigns aim to dismantle misconceptions and encourage earlier engagement with support systems.

[75] ([75]) further highlighted participants’ calls for increased availability of local health services, improved accessibility to medical consultations and interventions, and the importance of offering care in neutral, non-religious environments.

## 4. Discussion

This systematic review unequivocally highlights the central role of stigma in shaping attitudes and behaviors related to help-seeking among people with dementia, ultimately hindering timely access to diagnosis, adequate support, and appropriate therapeutic interventions. Stigma emerges as a pervasive factor, intricately interwoven with both individual dimensions, including psychological well-being and self-esteem, and contextual dynamics, including social relationships, media representations, and public policy frameworks.

One of the most significant contributions in this area is the study by [55] ([55]), the first to focus exclusively on the perspectives of people with dementia and their caregivers regarding the diagnostic process. Their findings underscore the extent to which fear associated with the diagnosis, negatively perceived prognosis, and apprehension about others’ reactions constitute major barriers to timely help-seeking. Our review further elaborates on this process, revealing an adaptive trajectory in response to the diagnosis and its progression, shaped by personal and societal representations of dementia.

Stigma emerges as an ambivalent construct. On the one hand, it fosters avoidance, isolation, and delays in seeking support ([12]; [7]; [13]; [47]; [72]; [83]); on the other hand, for some, receiving a diagnosis may bring relief and a sense of understanding ([48]). The findings of this review confirm previous literature ([55]; [76]) while extending the interpretative framework by demonstrating how the experience of stigma is influenced by the interaction between psychosocial and cultural variables.

Help-seeking is commonly defined as a deliberate and multi-stage process that includes the recognition of a problem, the identification of potential support resources, and the engagement in communication to access these resources ([60]; [76]). According to [76] ([76]), the help-seeking process unfolds through the following stages: problem recognition, identification of necessary resources, evaluation of available resources, and the disclosure of one’s condition and needs to others.

The Theory of Planned Behavior (TPB; [1]) offers a valuable framework for understanding the psychological determinants of help-seeking intentions, particularly in the context of early dementia. TPB posits that behavioral intention is shaped by the following three main components: attitude toward the behavior, which reflects the individual’s positive or negative evaluations of performing the behavior, derived from behavioral beliefs linking the action to potential outcomes ([2]; [1]); subjective norms, referring to the perceived social pressure from significant others to engage or not engage in the behavior; and perceived behavioral control, which contributes not only to intention but may also directly influence behavior, especially when it accurately reflects actual control ([3]). The TPB provides a robust theoretical framework to interpret help-seeking behaviors by highlighting how the combination of personal beliefs, perceived social expectations, and perceived control over one’s actions can drive or hinder engagement in such behaviors. In this context, TPB allows for a nuanced understanding of the interplay between stigma, personal attitudes, social influences, and perceived capacity to seek support.

Drawing on the conceptual framework of Corrigan and colleagues ([15]; [14]), stigma can be differentiated into the following two domains: public stigma, referring to society’s collective attribution of negative characteristics to a social group, and self-stigma, which describes the internalization of these stereotypes by the stigmatized individual, with detrimental consequences for self-esteem and quality-of-life ([42]; [61]; [18]).

According to [41] ([41]), the process of stigma internalization begins when individuals develop lay theories about mental illness from childhood conceptualizations that reflect cultural images. [41] ([41]) introduce the concept of stereotype awareness: awareness of general negative beliefs about mental illness propagated by one’s culture. Internalization of stigma starts with an agreement on stereotypes and then an endorsement of stereotypes perceived as common in the public. The process becomes harmful with self-concurrence, in which individuals believe that these culturally internalized beliefs apply to them ([41]). Stigma internalization is a phenomenon strictly related to anticipated discrimination ([64]). This component was found to constitute a second potential pathway through which discrimination may lead to decreased well-being. The model proposed by [65] ([65]) suggests that anticipation of further discrimination as a result of experienced discrimination is independent of levels of internalized stigma (stereotype endorsement and alienation).

Consistent with these theoretical frameworks, our review documents how people with dementia perceive societal stereotypes and discrimination, which negatively affect self-perception and contribute to emotional experiences such as sadness, anger, loneliness, and a sense of lost autonomy ([56]; [12]; [13]; [46]; [72]). Psychological well-being, particularly anxiety and the desire to preserve autonomy, is often associated with delays in seeking support ([51]). However, not all individuals internalize stigma to the same extent: active rejection of stereotypes may serve as a protective factor ([58]; [70]). Some individuals are aroused by the stigma and express righteous anger, whilst others do not experience a decrease in self-esteem or become justifiably angry ([16]).

Several factors seem crucial in the relationship between stigma and help-seeking. [68] ([68]) in their study highlighted how stigma may indirectly contribute to cognitive decline through mechanisms such as social isolation and reduced engagement in activities, while social participation may exert a protective effect. Consistent with our findings, the presence of social networks plays a key role in reducing social isolation and facilitating access to care ([56]; [29]; [40]; [46]; [72]; [79]).

Individuals associated with stigmatized people, such as family members, caregivers, and friends, may experience public stigmatization. This public perception of associates has been referred to as courtesy stigma ([15]). Similarly to the conceptualizations of stigma by [15] ([15]), [45] ([45]) conceptualized affiliate stigma. Caregivers, family members, and friends, being closely affiliated with a stigmatized individual, may be personally affected by the prevailing public stigma ([45]). These cognitive and affective effects of affiliate stigma may lead caregivers to conceal their status, withdraw from social relationships, or even distance themselves from the stigmatized individual to avoid association. In our review, the role of the caregiver is shown to be decisive in the help-seeking process ([47]). However, caregivers themselves may experience stigma ([15]; [45]), which can significantly impact their psychological well-being and the quality of care relationships ([37]; [58]).

This review also highlights the presence of stigma among healthcare professionals ([20]; [56]; [29]; [37]; [47]). Prejudice, misinformation, and stereotypical beliefs, often present even among general practitioners who are typically the first point of contact in the healthcare system, can hinder access to care and impair clinician–patient communication ([58]). The World Alzheimer Report ([4]) found that 65% of healthcare workers still consider dementia a normal part of aging.

Public representations of dementia, disseminated through media, campaigns, and preventive narratives, play a powerful role in constructing stigma ([20]; [46]; [47]; [75]; [83]). Visuals and messages that evoke fear, shame, or dehumanization reinforce negative perceptions and obstruct empathic identification with affected individuals ([50]). Furthermore, dominant representations emphasizing individual responsibility for prevention, when decontextualized, may exacerbate blame ([80]).

Against this backdrop, systemic interventions are essential. Awareness campaigns and educational programs promoting early diagnosis have shown efficacy ([43]). Nonetheless, only one in four countries has implemented national policies to support people with dementia and their caregivers ([81]), revealing a substantial gap in healthcare systems.

A recurrent theme emerging from participants across the reviewed studies was the perceived lack of alternatives or meaningful activities following diagnosis, highlighting the need for further investigation into this issue. In an effort to promote independent living within the community and delay institutionalization for as long as possible, the Netherlands has implemented the Meeting Centres Support Programme (MCSP) ([24], [21]). The MCSP is grounded in the adaptation-coping model ([8]; [22]), which aims to support people with dementia and their caregivers in managing the adaptive tasks required by the progression of the condition ([25]). These include strategies to enhance coping with disability, improve mood and behavior, and maintain a positive self-image.

The findings of this review, whilst thematically consistent across studies, must be interpreted in light of the sociocultural contexts in which the original research was conducted. Cultural norms, societal beliefs about aging and illness, and healthcare infrastructure significantly shape the way stigma is experienced and how help-seeking unfolds ([38]; [31]). For instance, reluctance to disclose a diagnosis in Chinese contexts was frequently linked to collectivist values and concerns about family reputation ([13]; [83]), while immigrant and minority populations in Western countries faced structural barriers compounded by cultural mismatches and mistrust in formal services ([12]; [7]). In rural or under-resourced areas, limited access to care and entrenched misconceptions about dementia often intensified stigma and delayed engagement ([75]; [35]). Although common patterns emerged—such as the negative impact of public stigma, the importance of caregiver support, and systemic service gaps—their specific manifestations varied by geographic and cultural setting. Therefore, while the core themes are likely relevant across diverse populations, the transferability of specific findings may be constrained in contexts that differ substantially from those represented in the included studies. Future research should further explore the cultural specificity of stigma and tailor interventions to the sociocultural realities of target populations.

The findings presented offer a complex and nuanced understanding of the dementia-related stigma phenomenon, indicating that the experience of stigma is deeply embedded in psychosocial, cultural, and institutional factors. The practical implications point to the urgent need to (1) strengthen the training of healthcare professionals; (2) promote culturally competent awareness campaigns; (3) support caregivers through targeted psychological and social interventions; and (4) reform public policies and ensure a more equitable and non-stigmatizing representation of dementia. Only through a multidimensional approach can timely access to care be facilitated, the quality-of-life for people with dementia be improved, and stigma be effectively addressed.

This systematic review exhibits several methodological strengths that enhance its contribution to the field. Adherence to the PRISMA and ENTREQ guidelines, application of the GRADE-CERQual approach, and prior protocol registration on PROSPERO reflect a high level of methodological rigor and transparency. Furthermore, the exclusive focus on the lived experiences of people with dementia addresses a significant gap in the existing literature. The international scope encompassing twelve countries provides valuable cross-cultural insights into the stigma–help-seeking relationship. However, certain limitations warrant consideration when interpreting the findings. The inclusion of solely qualitative studies, while offering rich experiential data, constrains the generalizability of results and precludes quantitative assessment of stigma’s impact on help-seeking behaviors. Additionally, despite the stated focus on people with dementia, the inclusion of studies with diverse participant profiles (i.e., caregivers, healthcare professionals) may dilute the centrality of the dementia perspective. Finally, the exclusive use of American spelling in the formulation of our research strategy may have led to the exclusion of studies employing British spelling conventions. Despite these limitations, this review’s integration of established theoretical frameworks and systematic thematic synthesis provides a valuable foundation for understanding the complex interplay between stigma and help-seeking behaviors in dementia, informing future interventions at individual, social, and institutional levels.

## 5. Conclusions

This systematic review demonstrates the pervasive influence of stigma on help-seeking behaviors in dementia, revealing intricate interactions between individual, social, and institutional factors. Our findings establish stigma as a critical barrier to timely diagnosis and appropriate care intervention, with ramifications extending beyond affected individuals to healthcare systems and broader communities.

The evidence synthesized herein necessitates multi-level interventions addressing both individual adaptation and the structural determinants that perpetuate stigma. While our centering of the lived experiences of people with dementia represents a methodological strength, the reliance on exclusively qualitative studies indicates a need for methodological diversification in subsequent research.

Future empirical investigations should prioritize the development and evaluation of culturally responsive interventions across diverse populations, particularly examining the efficacy of public education initiatives, caregiver support programs, and healthcare professional training. Cross-cultural comparative analyses would substantially enhance the evidence base for policy formulation.

Addressing stigma-related barriers remains essential for developing inclusive dementia care systems globally and improving quality-of-life outcomes. This review contributes to this imperative by elucidating stigma’s multi-level manifestations and identifying strategic intervention points. Progress requires interdisciplinary collaboration to dismantle stigmatizing practices and foster environments where people with dementia can access support without experiencing discrimination or marginalization.

## Figures and Tables

**Figure 1 behavsci-15-00774-f001:**
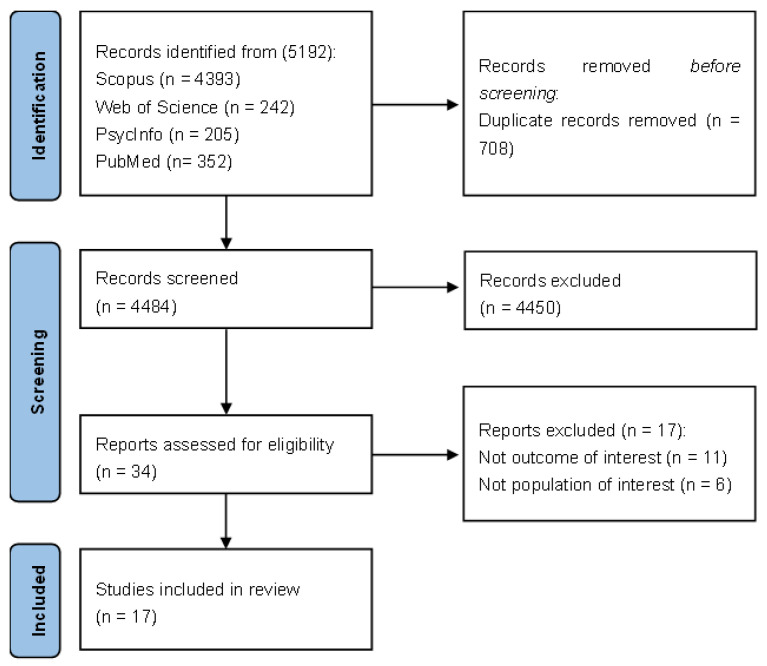
PRISMA flowchart.

**Table 1 behavsci-15-00774-t001:** Search strategy.

Search terms corresponding to ‘exposure’ included ‘stigma’, ‘stigmatization’, ‘self-stigma’, ‘internalized stigma’, ‘subjective stigma’, ‘public-stigma’, ‘social stigma’, and ‘societal stigma’, as well as terms related to help-seeking attitudes and behaviors, respectively, ‘help-seeking’, ‘treatment-seeking’, ‘care-seeking’, ‘healthcare utilization’, ‘seeking assistance’, ‘seeking support’, ‘seeking help’, ‘utilization of services’, and ‘barrier’. The basic search string based on Boolean operators was applied into each database as follow: [Dementia OR “Alzheimer’s disease” OR “Neurocognitive disorder” OR “Neurodegenerative disease”] AND [Stigma OR Stigmatization OR Self-stigma OR “Internalized stigma” OR “Subjective stigma” OR Public-stigma OR “Social stigma” OR “Societal stigma”] AND [Help-seeking OR Treatment-seeking OR Care-seeking OR “Healthcare utilization” OR “Seeking assistance” OR “Seeking support” OR “Seeking help” OR “Utilization of services” OR Barrier*].

*Note:* ‘*’ indicates truncation used to retrieve all terms that begin with the same root word (e.g., barrier* retrieves barrier, barriers, etc.).

**Table 2 behavsci-15-00774-t002:** The critical appraisal skills programme ([19]).

	*Section A: Are the Results Valid?*						*Section B: What Are the Results?*			*Section C: Will the Results Help Locally?*
	1. Was there a clear statement of the aims of the research?	2. Is a qualitative methodology appropriate?	3. Was the research design appropriate to address the aims of the research?	4. Was the recruitment strategy appropriate to the aims of the research?	5. Was the data collected in a way that addressed the research issue?	6. Has the relationship between researcher and participants been adequately considered?	7. Have the ethical issues been taken into consideration?	8. Was the data analysis sufficiently rigorous?	9. Is there a clear statement of findings?	10. How valuable is the research?
([7])	Yes	Yes	Yes	Yes	Yes	Cannot tell	Cannot tell	Yes	Yes	Yes
([12])	Yes	Yes	Yes	Cannot tell	Yes	Yes	Yes	Yes	Cannot tell	Yes
([13])	Yes	Yes	Yes	Yes	Yes	Yes	Yes	Yes	Yes	Yes
([20])	Yes	Yes	Yes	Yes	Yes	Yes	Yes	Yes	Yes	Yes
([26])	Yes	Yes	Yes	Yes	Yes	Yes	Yes	Yes	Yes	Yes
([29])	Yes	Yes	Yes	Yes	Yes	Yes	Yes	Yes	Yes	Yes
([37])	Yes	Yes	Yes	Yes	Yes	Yes	Cannot tell	Yes	Yes	Yes
([40])	Yes	Yes	Yes	Yes	Yes	Yes	Yes	Yes	Yes	Yes
([46])	Yes	Yes	Yes	Yes	Yes	Yes	Yes	Yes	Yes	Yes
([47])	Yes	Yes	Yes	Cannot tell	Yes	Yes	Yes	Yes	Yes	Yes
([48])	Yes	Yes	Yes	Yes	Yes	Yes	Yes	Yes	Yes	Yes
([56])	Yes	Yes	Yes	Yes	Yes	Yes	Yes	Yes	Yes	Yes
([70])	Yes	Yes	Yes	Cannot tell	Yes	Cannot tell	Cannot tell	Yes	Yes	Yes
([73])	Yes	Yes	Yes	Yes	Yes	Yes	Cannot tell	Yes	Yes	Yes
([75])	Yes	Yes	Yes	Cannot tell	Yes	Yes	Cannot tell	Yes	Yes	Yes
([79])	Yes	Yes	Yes	Yes	Yes	Yes	Cannot tell	Yes	Yes	Yes
([83])	Yes	Yes	Yes	Yes	Yes	Yes	Yes	Yes	Yes	Yes

**Table 3 behavsci-15-00774-t003:** Study characteristics.

Authors (Year) and Country	Aim	Study Design and Methodology	Sample	Key Results
**[7] ([7])** **Canada**	To address the factors that influence equitable access to and social participation in dementia care and support programs among foreign-born individuals including recent and non-recent immigrants and refugees	Qualitative—Interviews	Seven participants: two people with dementia, three caregivers, and two healthcare professionals	Foreign-born individuals face barriers in accessing dementia care Culturally inclusive activities can boost participation in programs Training HCPs in communication skills enhances cultural competence Raising awareness of dementia as a disease promotes timely care Increased awareness encourages migrants to seek help sooner
**[12] ([12])** **United Kingdom**	To explore how experiences of immigration and effects of globalization on family life and traditional kinships might explain delayed diagnosis and engagement with and by health services	Qualitative—Semi-structured interviews	A total of 61 participants: 10 people with dementia; 30 family members; 16 healthcare professionals; 2 interpreters; and 3 paid carers	Cultural and religious beliefs influence how dementia symptoms are recognized and when help is sought, especially in ethnically diverse communities Language barriers and interpretation challenges often delay diagnosis and create misunderstandings in care Stigma and mistrust in services lead families to present physical symptoms instead of cognitive concerns to avoid embarrassment Second-generation carers balance traditional expectations with Western healthcare norms, often facing family and cultural tensions Community context and migration history affect help-seeking behavior, with some individuals preferring to reconnect with their home country during symptom appraisal
**[13] ([13])** **New Zealand**	To explore their understanding of dementia and experiences of living with dementia	Qualitative—Semi-structured interviews	A total of 16 participants: 5 dyads of people with dementia and a family member, 6 people with dementia, and 5 family members of people with dementia	Participants referred to dementia as “brain shrinkage”, which linked it to aging and delayed recognition as a medical issue Families found symptoms like anhedonia and apathy to be particularly distressing Dementia literacy improved after diagnosis, mainly through online resources, but gaps in understanding still exist Caregiving involved tension between filial piety and emotional/physical stress, made worse by stigma Stigma acted as a barrier to care, but interventions like CST and psychoeducational programs helped reduce stress and improve support
**[20] ([20])** **United Kingdom**	(1) To find out about the experiences of current post-diagnosis support from people living with dementia (2) To explore ideas and suggestions of people living with dementia to improve post-diagnosis support	Qualitative—Focus groups	A total of 28 participants: 18 people with dementia and 10 spouses who were there to provide support	Post-diagnosis support is highly inconsistent across the UK Stigma around dementia leads to under-prioritization by professionals, leaving many feelings neglected compared to those with other conditions Peer and voluntary support are vital lifelines Financial struggles and navigating the benefits system are major challenges Co-production with people living with dementia improved research outcomes and enhances empowerment and service relevance
**[26] ([26])** **United States**	To discuss how the social construction of dementia shapes the health and social care interactions and experiences of people living with an AD/ADRD and their caregivers	Qualitative—Interviews	A total of 11 participants: 3 people with dementia and 8 caregivers	Perceptions of dementia influence the experiences Negative healthcare interactions include dismissal and dehumanization Negative perceptions perpetuate stigma and lower care quality Consequences include infantilizing language, diagnostic delays, and therapeutic nihilism
**[29] ([29])** **Australia**	To increase the knowledge of the needs of people living with dementia and those who provide informal or formal support to someone living with dementia in the Gippsland region	Qualitative—Interviews	A total of 26 participants: 25 people with dementia and 1 dyad	Participants identified unmet needs for people with dementia and caregivers Barriers included limited services, inadequate training, low dementia awareness, and systemic stigma Individuals struggled to navigate support, while providers lacked resources and training Social isolation and stigma worsened these challenges
**[37] ([37])** **India**	To understand attitudes and perceptions concerning people living with dementia residing in India from two diverse metropolitan cities, Delhi and Chennai	Qualitative—focus group discussions and individual interviews	A total of 58 participants: 15 from the public, 16 healthcare practitioners, 19 dementia carers, and 8 people with dementia	Significant knowledge gaps about dementia Cultural perceptions and stigmatizing terms like “lunatic” contributed to negative views and self-stigma Dementia was linked to religious beliefs and destiny Dementia care is viewed as a family responsibility
**[40] ([40])** **China**	To understand the experiences of people with dementia and their caregivers in engaging in dementia diagnosis	Qualitative—focus group discussions and individual interviews	A total of 18 people with dementia	Timely diagnosis is possible when individuals with memory problems are informed and empowered to seek help Lack of dementia services in primary care and insufficient education hinder progress A family-inclusive approach to educational interventions is needed Lack of social support services impacts the perception of dementia as untreatable
**[46] ([46])** **United Kingdom**	To explore barriers to self-management among people living with dementia	Qualitative—Interviews	A total of 11 participants: 7 people with dementia, 2 family members, and 2 charity representatives	Memory loss was a key barrier to dementia management Late diagnosis hindered self-management, highlighting the need for earlier detection Labeling family members as “carers” may disempower people with dementia Healthcare systems often promote dependence, focusing on caregivers instead of people with dementia Stigma exacerbates feelings of helplessness and exclusion
**[47] ([47])** **Northern Ireland**	To explore current public perceptions of living well with dementia from the perspective of people with dementia	Qualitative—Focus group	A total of 20 people with dementia	Job loss and driving restrictions after diagnosis reflect negative societal assumptions Positive public perceptions and media efforts’ promote inclusion and better attitudes Healthcare professionals often focus on end-stage dementia, making people feel passive in care decisions Participants preferred active, empowering approaches that maximize independence
**[48] ([48])** **Norway**	To explore the experience of living with cognitive impairment compatible with a possible dementia and the impact of being diagnosed with dementia	Qualitative—Semi-structured interviews	A total of 15 participants: 6 people with dementia, and 9 with cognitive impairment compatible with possible dementia (not diagnosed with dementia)	Diagnosis brought relief, clarity, and better understanding from loved ones Those without a diagnosis felt comfortable but faced social isolation Those diagnosed expressed concerns about dignity loss and the future Timely diagnosis was seen as crucial for managing the disease Maintaining control and self was important for participants with dementia
**[56] ([56])** **Australia**	What support was offered to people with dementia and their carers in Australian memory clinics in the year following the diagnosis? 2) What do people with dementia and carers think are the barriers and facilitators to accessing and utilizing post-diagnostic services? 3) What do people with dementia and carers think should be the ideal post-diagnostic support offered by memory clinics in the first year following diagnosis?	Qualitative—Semi-structured interviews	A total of 30 participants: 10 people with dementia; 13 caregivers of people with dementia; 3 people with young-onset dementia; and 4 caregivers of people with young-onset dementia	Memory clinics were overly focused on medication Stigma, grief, and emotional adjustment were major barriers to accessing services Family support was the most crucial facilitator Participants emphasized the need for a consistent, trusted point of contact Clear, tailored, and ongoing communication was essential
**[70] ([70])** **Israel**	To investigate the existence of identity and treatment stigma and if so, how stigmatic views influence on the presence of welfare stigma	Qualitative—Interviews and focus groups	A total of 50 participants: 10 people with dementia, 25 relatives, and 15 professionals	Participants felt self-stigma and invisibility Caregivers and professionals showed respect LTCI process felt disconnected and bureaucratic LTCI benefits were valued but rights were unclear Families faced high out-of-pocket costs
**[72] ([72])** **North Wales,** **United Kingdom**	(1) To explore the attitudes toward self-management held by people with early-stage dementia and their family caregivers; (2) to examine their views and perceptions of self-management and explored factors that could make self-management difficult	Qualitative—Semi-structured interviews	A total of 24 participants: 13 people with dementia, and 11 caregivers	Self-management was seen as coping and self-care Barriers included symptoms, low confidence, and stigma Caregivers focused on challenges, people with dementia valued independence Strategies included mental activity, positivity, and peer support
**[75] ([75])** **Tanzania**	To understand people with dementia and their caregivers’ experiences, identify challenges of living with dementia in Tanzania, and explore perceived support needs	Qualitative—Semi-structured interviews	A total of 26 participants: 14 people with dementia (2 of them were interviewed with their caregivers to support communication), and 12 caregivers	Women faced care barriers due to societal roles Symptoms included forgetfulness, aggression (men), and depression (women) Dementia was often misattributed to aging or witchcraft Education needed; radio and community spaces are most effective Caregivers felt overwhelmed with little formal support Herbal remedies were common; non-drug care preferred Co-morbidities and travel costs hindered access
**[79] ([79])** **Pakistan**	To explore respondents’ experiences with help-seeking, understandings of dementia, experiences with stigma, and the role of religion	Qualitative—Semi-structured interviews	A total of 20 people with dementia	Some had prior dementia awareness; others were diagnosed by professionals Dementia seen as aging reduced stigma Education and medical family ties aided understanding Low stigma overall and family frustration Dementia symptoms disrupted daily prayers
**[83] ([83])** **China**	To understand what challenges and tensions people with dementia and their family caregivers are facing in the context of dementia care services	Qualitative—Interviews	A total of 24 participants: 14 caregivers, and 10 people with dementia (2 participants were a care dyad)	Support services were lacking, causing exclusion Better financial, service, and psychological support needed Long-term care quality was poor Stigma increased isolation and discouraged help-seeking Post-diagnosis support was often absent

## Data Availability

No new data were created or analyzed in this study. Data sharing is not applicable to this article.

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
