# Peer review of "Within My Walls, I Escape Being Underestimated: A Systematic Review and Thematic Synthesis of Stigma and Help-Seeking in Dementia"

_behavsci, 2025, doi:10.3390/bs15060774_

Round 1

Reviewer 1 Report

Comments and Suggestions for Authors

Comments on the Quality of English Language

Additionally, I recommend re-reading the manuscript to ensure that the English is clear and consistent throughout.

Author Response

Abstract

Comment 1: Can you clarify whose help-seeking behaviours this review is targeting? This is not currently clear.

Response 1: Thank you for your comment. The abstract has been carefully revised, and the target of our review has been specified as follows:

Please, see p. 1, lines 14 – 16: “This review examines the impact of stigma on help-seeking among people with dementia and identifies key factors influencing this relationship.”

Comment  2: The abstract mentions that you searched qualitative and quantitative studies, but the title suggests you only included qualitative studies. I would recommend removing the types of studies here, but it is important to clarify why quantitative studies were also searched in the methods section.

Response 2: Thank you for your comment. The title has been changed for the purpose of better understanding. As suggested, the rationale for not including quantitative studies has been clarified in the method section.

Please, see p. 4, lines 153 – 155: “Following the screening process, no eligible quantitative studies were identified; therefore, only qualitative studies meeting the inclusion criteria were incorporated into the final synthesis.”

Comment 3: Please specify which method was used to analyse the data (e.g., thematic synthesis).

Response 3: Thank you for your comment. The method was specified in the abstract.

Please, see p. 1, line 20: “Thematic synthesis was employed to analyze the findings.”

Introduction

Comment  4: The second paragraph defines stigma well, but you mention 'prejudice', 'discrimination', and 'stereotypes' without defining these terms. It would strengthen the paragraph to briefly define them. Additionally, the transition to self-stigma feels abrupt, as it appears in a later paragraph. Consider integrating this paper with earlier definitions.

Response 4: Thank you for your comment. The introduction has been reworded and reorganized and has been updated with more appropriate bibliographical references.

Please, see p. 2, lines 64 – 70: “This framework further conceptualizes stigma in terms of stereotypes, prejudice, and discrimination: stereotypes involve generalized negative beliefs about a group, individuals with a specific condition, or oneself; prejudice is the emotional endorsement of these stereotypes, often expressed through negative emotional reactions toward others or oneself; and discrimination represents the behavioral response to prejudice, including exclusionary behaviors and diminished willingness to provide support (Werner et al., 2024; Nguyen & Li, 2020; Young et al., 2019).”

Comment 5: Several paragraphs (e.g., paragraph 3) lack references or would benefit from more recent citations. Specifically, papers mentioned in paragraph 4 could be strengthened by incorporating newer references.

Response 5: Thank you for your comment. The indication was precious, which is why a reorganization of the entire section was made.

Comment  6: While you mention Parker et al.'s review, your paragraph could benefit from more details about how your review builds on theirs, or what their review lacked and why your review is necessary.

Response 6: Thank you for your comment. Starting with the review by Parker et al. (2020), the authors' findings were indicated, and at the same time it was indicated how our review aims to go a step further to shed more light on the topic.

Please, see p. 3 , lines 107 - 115: “Parker et al. (2020) attempted to systematize the factors that hinder and facilitate the diagnostic process. Their findings indicate that stigma significantly impacts physical, psychological, and social well-being, and can also act as a barrier to research participation, as evidenced by recruitment challenges. However, the review by Parker et al. (2020) does not focus solely on the experiences of people with dementia, but also includes caregiver perspectives, who, according to the study, constituted the majority of participants. For this reason, our review aims to make a step further and consolidate existing evidence on the relationship between stigma and help-seeking in people with dementia, with an exclusive focus on the lived experiences of people with dementia.”

Comment 7: You mention "thematic analysis," but since this is a review, it seems you conducted a "thematic synthesis." Please clarify this terminology.

Response 7: Thank you for your comment. The change has been made to the entire manuscript.

Comment 8 :Before presenting your research questions, you introduce terms like "resilience" without explaining them. These terms could seem speculative without prior mention, and it may be helpful to establish their relevance earlier.

Response 8: Thank you for the comment. It was decided to remove the term “resilience” as it is not used in the rest of the manuscript. Therefore, it was chosen to replace it.

Please, see p. 3, lines 116 – 119: “The primary aim is to clarify the relationship between stigma and help-seeking, with particular attention to the factors that influence this connection. These may include individual characteristics (e.g., self-esteem, self-efficacy, cultural values) and social factors (e.g., support networks, community attitudes).”

Comment 9: You refer to factors that "modulate" the relationship between stigma and help-seeking. This phrasing implies an analysis of effect sizes, which is more typical of quantitative research. Please rephrase to make this more appropriate for a qualitative review. The same applies to your results section.

Response 9: Thank you for your comment. As has been pointed out, the term “modulate” refers to a type of quantitative research. For this reason, it was preferred to use terms such as “impact” and “influence.”

Methods

Comment 10: It's great that you're using a framework for your search terms. Could you provide a brief explanation of what PICO means for readers who may not be familiar with it?

Response 10: Thank you for your comment. As suggested, a brief explanation of what the PICO framwork indicates has been provided.

Please, see p. 3, lines 127 – 133: “The search strategy was developed using the PICO framework, a widely adopted tool in medical and systematic review research that facilitates the formulation of structured and well-defined research questions, thereby aiding the identification and evaluation of relevant evidence (Higgins, 2008). In the present review, the PICO framework was applied to define the scope of the investigation, focusing on: people with dementia (Population); their exposure to stigma (Interest); no comparator was included (Comparison); and the outcomes related to help-seeking attitudes, behaviors, and service utilization (Outcome).”

Comment 11: While your search included both qualitative and quantitative studies, you only included qualitative studies in the final review. Can you clarify why quantitative studies were excluded?

Response 11: Thank you for the comment. As suggested, the rationale for not including quantitative studies has been clarified in the method section.

Please, see p. 4, lines 153 – 155: “Following the screening process, no eligible quantitative studies were identified; therefore, only qualitative studies meeting the inclusion criteria were incorporated into the final synthesis.”

Comment 12: Your last search was conducted on 15 April 2024. I recommend updating your search to ensure the most current studies are included.

Response 12: Thank you for your comment. As suggested, an update to the review has been made.

Comment 13: Did you use truncation and wildcards for your search terms? For example, your search suggests an emphasis on American spelling ("stigmatization"), but it might exclude British spelling ("stigmatisation"). Please clarify this.

Response 13: Thank you for your comment. In formulating the research strategy, more emphasis was placed on the American spelling given the greater presence of bibliographic references. Nevertheless, this choice is a limitation of the research.

Please, see p. 18, lines 621 – 623: “Finally, the exclusive use of American spelling in the formulation of our research strategy may have led to the exclusion of studies employing British spelling conventions.”

Results

Comment 14: When describing study characteristics, could you combine "interviews" and "semi-structured interviews"? Or do you mean "structured interviews" by "interviews"? This needs clarification.

Response 14: Thank you for your comment. As suggested, “interviews” and “semi-structured interviews” have been combined. In order to clarify the ambiguity between “structured interviews” and “interviews,” I emphasize that among the included studies, structured interview is not mentioned as an instrument.

Please, see p. 7, lines 274 – 275: “The qualitative methods employed included individual interviews and semi-structured interviews (n = 12), …”

Comment 15: The title for Table 3 is missing. Additionally, the presentation of key findings is quite lengthy. I recommend shortening key findings in the table, perhaps using bullet points for easier reading.

Response 15: Thank you for your comment. As suggested, the title of Table 3 has been inserted and the key findings column of Table 3 has been reduced.

Comment 16: When introducing the themes, you use the phrase "main issues." I recommend changing this to something more neutral and objective, such as "key themes" or "key factors."

Response 16: Thank you for your comment. As suggested, the term “key themes” was used instead of “main issues.”

Please, see p. 12, line 289: “Five key themes …”

Please, see p. 14, line 382 : “Four key themes …”

Comment 17: Did you distinguish between data from people with dementia and data from carers? If not, could you explain why this distinction wasn't made? It’s not clear from the methods or results.

Response 17: Thank you for the comment. During all stages of the review, we aimed to include relevant information gathered directly from people with dementia and excluded different perspectives. In order to make this clearer, it has been specified in the text.

Please, see p. 7, lines 264 – 267: “Studies that did not explicitly indicate the involvement of people with dementia or where it was unclear whether their perspectives were solely represented were excluded. Collectively, these studies underscore the pervasive nature of dementia-related stigma and its substantial influence on help-seeking behaviors and pathways.”

Comment 18: Some studies, such as Molvik et al. (2024), focus on living with the diagnosis and selfdisclosure/concealment rather than help-seeking behaviours. Could you clarify how these studies contribute to understanding help-seeking? Additionally, since self-disclosure and stigma can be related to help-seeking, which you suggest by including this study, it is worth considering including more studies on this topic (e.g., Kohl et al., "It’s just geeng the word out there: Selfdisclosure by people with young-onset dementia").

Response 18: Thank you for your comment. In the case of studies such as that of Molvik et al. (202) addressing issues such as self-disclosure/concealment were included because in the theoretical references used in our background, these aspects influence the help-seeking process even following diagnosis (Ajzen, 1991; Ajzen & Kruglaski, 2019; Werner et al., 2014). According to the literature, stigma is negatively correlated with help-seeking (Carter et al., 2024; Dooley et al., 2025). The presence of a support network, reduced isolation and thus lower levels of stigma may lead to disclosure and coping with “treatment” more smoothly (Lian et al., 2017; Walker et al., 2023).

With respect to the suggested study by Kohl et al. (2024), it was excluded because the sample included people with dementia under age 60, going against the criteria chosen for our review.

Discussion

Comment 19: While you describe your findings in detail, comparing them to the existing literature could strengthen your discussion by situating your results in the broader context.

Response 19: Thank you for the comment. Based on the suggested directions, the “Discussion” section has been reorganized to incorporate the comparison with the most recent literature.

Comment 20: I agree that interventions and campaigns are needed to tackle dementia-related stigma. Could you provide more specific suggestions for such efforts based on the findings from your review?

Response 20: Thank you for the comment. Based on the results and the discussion developed, more specific suggestions were given.

Plaese, see p. 18, lines 601 - 609:” The findings presented offer a complex and nuanced understanding of the dementia-related stigma phenomenon, indicating that the experience of stigma is deeply embedded in psychosocial, cultural, and institutional factors. The practical implications point to the urgent need to: (1) strengthen the training of healthcare professionals; (2) promote culturally competent awareness campaigns; (3) support caregivers through targeted psychological and social interventions; and (4) reform public policies and ensure a more equitable and non-stigmatizing representation of dementia. Only through a multidimensional approach can timely access to care be facilitated, the quality of life for people with dementia be improved, and stigma be effectively addressed.

Comment 21: I agree that a strength of your review is its focus on studies that include people with dementia. However, didn’t Parker et al. also focus on this? It would be helpful to explain whether there are any specific strengths in focusing exclusively on qualitative studies.

Response 21: Thank you for the comment. As mentioned in the introduction section, Parker et al. (2020) also focus on the experiences of people with dementia but included caregivers of people with dementia in the study, not allowing for specific analysis of the perspective of people with dementia.

With respect to the strengths related to having analyzed only qualitative studies, several aspects were included.

Please, see p. 18, lines 612 - 615: ”… the exclusive focus on the lived experiences of people with dementia addresses a significant gap in existing literature. The international scope encompassing twelve countries provides valuable cross-cultural insights into the stigma-help seeking relationship.”

Comment 22: Your limitations section could be strengthened. I recommend providing more detail on how the limitations you identified impact the findings and their generalizability.

Response 22: Thank you for the comment. As suggested, the comments section has been improved.

Please, see p. 18, lines 615 - 620: “However, certain limitations warrant consideration when interpreting the findings. The inclusion of solely qualitative studies, while offering rich experiential data, constrains the generalizability of results and precludes quantitative assessment of stigma's impact on help-seeking behaviors. Additionally, despite the stated focus on people with dementia, the inclusion of studies with diverse participant profiles (i.e. caregivers, healthcare professionals) may dilute the centrality of the dementia perspective.”

Comment 23: The conclusion could benefit from being more specific and grounded in your findings. Rather than summarizing the results in general terms, consider highlighting the key mechanisms and contextual factors identified through your synthesis. Additionally, future research directions should be clearly grounded in the gaps revealed by the review. It would also be helpful to briefly highlight the practical implications of your results for intervention development.

Response 23: Thank you for the comment. As suggested, there has been a reorganization of the conclusion with the goal of being more specific and more grounded in our findings.

Please, see p. 18, lines 627 – 647: “This systematic review demonstrates the pervasive influence of stigma on help-seeking behaviors in dementia, revealing intricate interactions between individu-al, social, and institutional factors. Our findings establish stigma as a critical barrier to timely diagnosis and appropriate care intervention, with ramifications extending be-yond affected individuals to healthcare systems and broader communities.

The evidence synthesized herein necessitates multi-level interventions addressing both individual adaptation and the structural determinants that perpetuate stigma. While our centering of the lived experiences of people with dementia represents a methodological strength, the reliance on exclusively qualitative studies indicates a need for methodological diversification in subsequent research.

Future empirical investigations should prioritize the development and evaluation of culturally responsive interventions across diverse populations, particularly exam-ining the efficacy of public education initiatives, caregiver support programs, and healthcare professional training. Cross-cultural comparative analyses would substan-tially enhance the evidence base for policy formulation.

Addressing stigma-related barriers remains essential for developing inclusive dementia care systems globally and improving quality of life outcomes. This review contributes to this imperative by elucidating stigma's multi-level manifestations and identifying strategic intervention points. Progress requires interdisciplinary collabora-tion to dismantle stigmatizing practices and foster environments where people with dementia can access support without experiencing discrimination or marginalization.” 

Reviewer 2 Report

Comments and Suggestions for Authors

Author Response

Abstract:

Comment 1: The opening sentence could be refined for clarity and impact and clarity

Response 1: Thank you for the comment. The opening sentence has been reorganized.

Please, see p. 1, lines 13 – 16: “Dementia-stigma related significantly influence help-seeking and affect the quality of care and support received by people with the condition. This review examines the impact of stigma on help-seeking among people with dementia and identifies key factors influencing this relationship.”

Comment 2: Please clarify inclusion criteria

Response 2: Thank you for your comment. As suggested, the inclusion criteria have been clarified.

Please, see p. 1, lines 18 – 21: “… that met the inclusion criteria. These criteria encompassed studies focused on individuals aged 60 and older, addressing public or self-stigma, and exploring help-seeking behaviors and related influencing factors. A thematic synthesis was employed to analyze the findings.”

Comment 3: Please briefly describe how themes were identified

Response 3: Thank you for your comment. How themes were identified is specified in the abstract.

Please, see p. 1, line 20: “Thematic synthesis was employed to analyze the findings.”

Introduction:

Comment 4: The introduction is quite repetitive, would it be possible to reduce this by removing some of the repeated ideas that are appearing multiple please (stigma delaying diagnosis/help-seeking, public vs, self-stigma). consolidating these points would improve flow and readability.

Response 4: Thank you for your comment. The introduction has been reorganized and reorganized in order to improve flow and readability.

Comment 5: Second paragraph: ‘an undesirable stereotypes’ should be stereotype - Second paragraph ‘is dependent by’ should be ‘dependent on’ - Is it accurate to say it remains untreated?

Response 5: Thank you for your comment. In reformulating the text, in order to incorporate the new information and in making the text more fluent, this sentence has been modified.

Drawing on the definition used by Bacsu et al. (2024), which complements and expands on the one provided by Goffman (1963), it is possible to state that dementia remains untreated because of unfair and unequal treatment.

Please, see p. 2, lines 46 - 48: “Stigma has been defined as a socially discrediting attribute or characteristic that encompasses negative beliefs, insufficient knowledge, and discriminatory behaviors, ultimately leading to unjust or unequal treatment (Bacsu et al., 2022; Goffman, 1963).”

Comment 6: You claim that “no study has exclusively focused” on the perspective of individuals with dementia. However, you later cite Parker et al. (2020) as a review including people with dementia and caregivers. Be more precise about your unique contribution—e.g., “While Parker et al. considered both people with dementia and caregivers, our review focuses specifically on…

Response 6: Thank you for your comment. Starting with the review by Parker et al. (2020), the authors' findings were indicated, and at the same time it was indicated how our review aims to go a step further to shed more light on the topic.

Please, see p. 3 , lines 107 - 115: “Parker et al. (2020) attempted to systematize the factors that hinder and facilitate the diagnostic process. Their findings indicate that stigma significantly impacts physical, psychological, and social well-being, and can also act as a barrier to research participation, as evidenced by recruitment challenges. However, the review by Parker et al. (2020) does not focus solely on the experiences of people with dementia, but also includes caregiver perspectives, who, according to the study, constituted the majority of participants. For this reason, our review aims to make a step further and consolidate existing evidence on the relationship between stigma and help-seeking in people with dementia, with an exclusive focus on the lived experiences of people with dementia.”

Comment 7: The last two paragraphs o the introduction could be combined for clarity and flow.

Response 7: Thank you for your comment. The introduction has been reorganized and reorganized in order to improve flow and readability. In the reorganization, the last two paragraphs have also undergone significant changes.

Comment 8: Rather than ‘addressing this knowledge gap…’ you could say this systematic review aimed (since you didn’t have a gap identified prior to that statement).

Response 8: Thank you for your comment. The introduction has been reorganized and reorganized in order to improve flow and readability. In the reorganization, the last two paragraphs have also undergone significant changes.

Please, see p. 3, lines 113 - 115: “Our review aims to make a step further and consolidate existing evidence on the relationship between stigma and help-seeking in people with dementia, with an exclusive focus on the lived experiences of people with dementia.”

Comment 9: The introduction could benefit from a reorganization to reduce redundancy and improve clarity.

Response 9: Thank you for your comment. The introduction has been reorganized and reorganized in order to improve flow and readability.

Methods:

Comment 10: Well-structured!

Response 10: Thank you for your comment.

Comment 11: Why qualitative only?

Response 11: Thank you for the comment. As suggested, the rationale for not including quantitative studies has been clarified in the method section.

Please, see p. 4, lines 153 – 155: “Following the screening process, no eligible quantitative studies were identified; therefore, only qualitative studies meeting the inclusion criteria were incorporated into the final synthesis.”

Comment 11: analysis can you provide examples of how you used the three stages? i.e., what were some of the things that came out of the initial coding for relevant themes? Same for descriptive…

Response 11: Thank you for your comment. Examples related to the three stages of the process have been proposed in the method section.

Please, see p. 5, lines 190 – 198: “The synthesis process followed three stages: (1) line-by-line coding of study findings, where each line of text was inductively coded by assigning descriptive labels to units of meaning (e.g., “Understanding of dementia”, “Impact on the person living with dementia”); (2) development of descriptive themes, in which similar codes were grouped together to reflect patterns across studies (e.g., “Lack of awareness”, “Perceived stigma”); (3) generation of analytical themes, where interpretations extended beyond the primary data to address the review question and offer new conceptual insights (e.g., “Reluctance to Share One’s Condition and Risk of Labelling”, “From Public Perception to Internalization or Rejection of Stigmatizing Beliefs”). No software was used during the process.”

Results:

Comment 12: Reluctance to share… strong theme! Can you consider clarifying if this reluctance stemmed more from anticipated stigma or from past experiences (otherwise, I am wondering how this is different from 3.2.1.2)

Response 12: Thank you for the comment. Through a reformulation of the result and thanks to the updated review, it is possible to infer that reluctance to share one's condition is rooted in phenomenon of preemptive avoidance of social situations (Pavković et al., 2025) and thus differentiable from the experience of internalizing stigma (Schauman et al., 2019).

Please, see p. 12, lines 298 – 301: “Individuals living with dementia expressed feelings of shame and insecurity linked to cognitive changes, which intensified their sense of self-doubt and fear of rejection, ultimately leading to anticipatory avoidance of social situations (Pavković et al., 2025).”

Please, see p. 17, lines 535 - 540: “Stigma internalization is a phenomenon strictly related to anticipated discrimination (Rüsch et al., 2006.) This component was found to constitute a second potential pathway through which discrimination may lead to decreased well-being. The model proposed by Schauman et al. (2019) suggests that anticipation of further discrimination as a result of experienced discrimination is independent of levels of internalized stigma (stereotype endorsement and alienation).”

Comment 13: Role of Family and community: Some of this is repetitive, can you consolidate similar findings? i.e., fear of social judgment appears in multiple studies; synthesize rather than recount

Response 13: Thank you for the comment. The theme that emerged was reorganized in order to consolidate similar studies and avoid redundancies.

Please, see p. 12, lines 325 – 337: “Many participants reported that stigmatizing attitudes were present within both familial and community contexts. These were often rooted in misconceptions about dementia or suspicions that individuals were exaggerating or fabricating symptoms, which delayed the initiation of help-seeking pathways (Hurzuk et al., 2022; Zhang et al., 2020). In one case, a participant described being mocked by a community member (Willis et al., 2020), while others recounted experiences of stress induced by communal negativity (Lian et al., 2017). These experiences often led to denial of diagnosis and further isolation, exacerbated by relatives’ fears of social judgment (Hurzuk et al., 2022).

However, some studies also reported supportive and empathetic family environments, with such attitudes often associated with higher educational attainment (Willis et al., 2020). These individuals, family members and community members, have the potential to support people with dementia in overcoming societal challenges or may inadvertently reinforce societal stigma surrounding dementia (Mitchell et al., 2020).”

Discussion:

Comment 14: Situating these findings within a theoretical framework would strengthen the relevance and deepen interpretation. Particularly when discussing the ambivalent role of stigma.

Response 14: Thank you for the comment. As suggested, some frameworks have been included that can help to understand how the relationship between stigma and help-seeking in people with dementia is expressed and how the factors that influence this relationship act.

Please, see p. 16, lines 501 – 540: “Help-seeking is commonly defined as a deliberate and multi-stage process that includes the recognition of a problem, the identification of potential support resources, and the engagement in communication to access these resources (Rickwood & Thomas, 2008; Werner et al., 2014). According to Werner et al. (2014), the help-seeking process unfolds through several stages: problem recognition, identification of necessary resources, evaluation of available resources, and the disclosure of one’s condition and needs to others.

The Theory of Planned Behavior (TPB; Ajzen, 1991) offers a valuable framework for understanding the psychological determinants of help-seeking intentions, particularly in the context of early dementia (...). TPB posits that behavioral intention is shaped by three main components: attitude toward the behavior, which reflects the individual’s positive or negative evaluations of performing the behavior, derived from behavioral beliefs linking the action to potential outcomes (Ajzen & Fishbein, 1980); subjective norms, referring to the perceived social pressure from significant others to engage or not engage in the behavior; and perceived behavioral control, which contributes not only to intention but may also directly influence behavior, especially when it accurately reflects actual control (Ajzen & Kruglanski, 2019). The TPB provides a robust theoretical framework to interpret help-seeking behaviors by highlighting how the combination of personal beliefs, perceived social expectations, and perceived control over one’s actions can drive or hinder engagement in such behaviors. In this context, TPB allows for a nuanced understanding of the interplay between stigma, personal attitudes, social influences, and perceived capacity to seek support.

Drawing on the conceptual framework of Corrigan and colleagues (Corrigan et al., 2005; Corrigan & Fong, 2014), stigma can be differentiated into two domains: public stigma, referring to society’s collective attribution of negative characteristics to a social group, and self-stigma, which describes the internalization of these stereotypes by the stigmatized individual, with detrimental consequences for self-esteem and quality of life (Lion et al., 2019; Riley et al., 2014; Corrigan et al., 2006).

According to Link and Phelan (2001), the process of stigma internalization begins when individuals develop lay theories about mental illness from childhood conceptualizations that reflect cultural images. Link & Phelan (2001), introduce the concept of stereotype awareness: awareness of general negative beliefs about mental illness propagated by one's culture. Internalization of stigma starts with agreement on stereotypes and then endorsement of stereotypes perceived as common in the public. The process becomes harmful with self-concurrence, in which individuals believe that these culturally internalized beliefs apply to them (Link & Phelan, 2001). Stigma internalization is a phenomenon strictly related to anticipated discrimination (Rüsch et al., 2006.) This component was found to constitute a second potential pathway through which discrimination may lead to decreased well-being. The model proposed by Schauman et al. (2019) suggests that anticipation of further discrimination as a result of experienced discrimination is independent of levels of internalized stigma (stereotype endorsement and alienation).

Conclusion:

Comment 15: Please briefly list the ‘mechanisms through which stigma operates’ - The impact statement and call to action could be strengthened.

    • Multi-level interventions supporting individual adaptation to diagnosis but also address the social and systemic structures that perpetuate stigma.
    • further empirical research is essential for designing culturally responsive, equity-focused, and contextually grounded interventions. Future studies should explore the impact of public education, caregiver support, and health system training in reducing stigma and promoting early engagement with care, especially among underserved and diverse populations.
    • Overcoming stigma-related barriers is vital for fostering more inclusive dementia care systems (which globally we are all aspiring to) and improving the quality of life for individuals affected by the condition.

Response 15: Thank you for your comment. Starting from the suggestions provided, the conclusion has been reorganized.

Please, see p. 18, lines 629 – 649: “This systematic review demonstrates the pervasive influence of stigma on help-seeking behaviors in dementia, revealing intricate interactions between individual, social, and institutional factors. Our findings establish stigma as a critical barrier to timely diagnosis and appropriate care intervention, with ramifications extending beyond affected individuals to healthcare systems and broader communities.

The evidence synthesized herein necessitates multi-level interventions addressing both individual adaptation and the structural determinants that perpetuate stigma. While our centering of the lived experiences of people with dementia represents a methodological strength, the reliance on exclusively qualitative studies indicates a need for methodological diversification in subsequent research.

Future empirical investigations should prioritize the development and evaluation of culturally responsive interventions across diverse populations, particularly examining the efficacy of public education initiatives, caregiver support programs, and healthcare professional training. Cross-cultural comparative analyses would substantially enhance the evidence base for policy formulation.

Addressing stigma-related barriers remains essential for developing inclusive dementia care systems globally and improving quality of life outcomes. This review contributes to this imperative by elucidating stigma's multi-level manifestations and identifying strategic intervention points. Progress requires interdisciplinary collaboration to dismantle stigmatizing practices and foster environments where people with dementia can access support without experiencing discrimination or marginalization.”

Reviewer 3 Report

Comments and Suggestions for Authors

Title: Internalization of Stigma and Public Stigma in Dementia and Help-Seeking: A Systematic Review of Qualitative Studies 

Journal: Behavioral Sciences 

Overall comment 

I thank the editor for the opportunity to review this manuscript which provides an important synthesis focused exclusively on people with dementia's and caregivers' experiences. The authors are commended for this much-needed approach on the topic of stigma and the impact on help seeking amongst a vulnerable population group. The study is methodologically rigorous and includes well-structured findings with translational relevance. Having said that, the manuscript requires a minor revision for grammatical accuracy, particularly of the introduction section. While the other sections are comprehensive, the Introduction also requires some revisions to broaden references to the relevant literature, to eliminate repetitions and to improve conciseness. 

Detailed suggestions for revisions are provided below. 

Title 

  • The use of wording in the title “Internalization of Stigma and Public Stigma in Dementia..." appears to be confusing some of the constructs you later define in the introduction e.g. I am not sure what “stigma” is being referred to as distinguished from “public stigma”. Are you referring to self-stigma which involves an internalisation of public stigma? Consider rewording and perhaps flagging the use of narrative synthesis. 

Abstract 

  • The Abstract is well written. Consider also including a mention of the analytic approach  

Introduction 

  1. Page 2, Paragraph 2; The statement, “Dementia is still highly stigmatized in modern societies” needs to be supported e.g., through referencing the appropriate research. Similarly, the statement “Individuals' knowledge and understanding of dementia significantly influence their help-seeking behaviors” should be supported by citation/s.
  2. Check the introduction for other statements requiring citations. For example, Page 2, Paragraph 5; “These inequalities can influence public policy and the allocation of resources for dementia care (Reference?). Negative perceptions and widespread misinformation about dementia contribute to a lack of funding for essential services and support systems (Reference?). This underinvestment exacerbates existing inequalities, leaving people with dementia and their carers without the necessary resources to manage the condition effectively (Reference?)”
  3. Similarly, some of the source article/s that illustrate the negative impact of stigma on dementia care (for PWD and Caregivers) are missing, and/or limited to the ADI report. You could, for example, briefly compare findings from ADI report with other findings. Also, see Brookman et al., 2025 for a detailed background on the literature relating to dementia-related stigma. 
  4. Page 3, paragraph 3, it is stated that, “To counteract this cycle, it is crucial to address both public and structural stigma.” However, structural stigma has not been defined. Please delete or include this earlier in the introduction when defining stigma.
  5. Overall, the introduction contains relevant content, but it is very lengthy, and repetitive. In addition, some of the content circles back on itself e.g., the relationship between dementia-related (DR) stigma and delayed diagnosis is mentioned multiple times in different ways and places. Revise to eliminate repetitions and consolidate ideas to improve conciseness, grammar and flow. For example, consider having a broad structure that takes the reader on a journey from a broad topic area (DR-stigma), to the narrower problem of the research gap (impact of DR-stigma on help seeking.; the global impact of dementia, definitions, public and self-stigma, the impact of dementia-related stigma on help seeking, and finally, current research aims.  
  6. Relatedly, there are some awkward transitions between ideas. Try and keep paragraphs and topic areas clearly defined, with transition sentences to the next idea. If the journal’s formatting guidelines permit it, subheadings can be helpful (written or imaginary).

Minor points 

  • The introduction needs a thorough edit and checks for grammatical errors (e.g., "dependent by" should be "dependent on,") and typographical error (e.g., “an individuals”)
  • ADI needs to be first spelt out first (ie. Alzheimer’s Disease International [ADI], 2024) prior to using their acronym. 

Methods 

  1. Thematic synthesis is a good choice of analysis for qualitative studies. However, further detail is required e.g., was a deductive or inductive approach taken? Also, was open coding and software (e.g., NVIVO) used?
  2. For transparency, why did the final review only include qualitative studies? As this is mentioned later in the Discussion section as a limitation, it would seem relevant to mention why the decision was made to only include qualitative studies. 

Minor points 

  • The content in the Method section is appropriate and well structured. A thorough grammar check would improve readibility. 

Results  

  1. The results section is comprehensive and with key components and themes and subthemes are well organised, and supported with relevant study citations. However, as with previous sections, revisions could be made to improve conciseness and grammar e.g., there is some switching between tense when reporting the findings.
  2. A minor formatting point: the key table in the Results sections covers multiple pages (pages 9-15). If the width of the initial columns were narrowed, and the final column Headed “Key results” was expanded, the table would be shortened and easier to read. 

Discussion 

  1. The discussion section is well written and relates well to the existing literature. There were occasional exceptions to this that require minor amendments to improve accuracy e.g. Page 20, paragragh 2, “Literature indicates that stigmatizing attitudes are more pronounced among individuals with limited knowledge of AD, those with little contact with people with dementia, men, younger individuals, and those influenced by cultural interpretations of dementia (Hermann et al., 2018).” This statement is not entirely accurate and needs amending, as the literature indicates that the relationship between public DR-stigma and age is inconsistent. For example, there is a systematic review that has arrived at a different conclusion e.g., that DR-stigma is less pronounced in younger cohorts (see Nguyen and Li, 2020). Also, Brookman et al., 2025 found that dementia knowledge was only related to increased stigma in younger (not older) adults.
  2. As mentioned in the Method section, it would seem relevant to mention why the decision was made to only include qualitative studies in the Method section, to then mention this as a limitation. 

Author Response

Title 

Comment 1: The use of wording in the title “Internalization of Stigma and Public Stigma in Dementia..." appears to be confusing some of the constructs you later define in the introduction e.g. I am not sure what “stigma” is being referred to as distinguished from “public stigma”. Are you referring to self-stigma which involves an internalisation of public stigma? Consider rewording and perhaps flagging the use of narrative synthesis.

Response 1: Thank you for your comment. In order to avoid confusion with respect to the terminologies chosen, the title has been reworded.

Please, see p. 1, lines 2 – 3: “I Cannot be Underestimated if I Stay at Home”: A Systematic Review and Thematic Synthesis” 

Abstract 

Comment 2: The Abstract is well written. Consider also including a mention of the analytic approach  

Response 2: Thank you for the comment. As suggested, the type of approach used was included in the abstract.

Please see p. 1, lines 20-21: “A thematic synthesis was employed to analyze the results.”

Introduction 

Comment 3: Page 2, Paragraph 2; The statement, “Dementia is still highly stigmatized in modern societies” needs to be supported e.g., through referencing the appropriate research. Similarly, the statement “Individuals' knowledge and understanding of dementia significantly influence their help-seeking behaviors” should be supported by citation/s.

Response 3: Thank you for the comment. In the rewording of the text this element was integrated into the text differently.

Please, see p. 1, lines 40 – 43: “Stigma can adversely affect the lives of those living with dementia and their close ones in various ways, including the induction of shame, which can deter individuals from seeking medical assistance or participating in research (Putland & Brookes, 2024; Siette et al., 2023; O’Connor et al., 2022).”

Comment 4: Check the introduction for other statements requiring citations. For example, Page 2, Paragraph 5; “These inequalities can influence public policy and the allocation of resources for dementia care (Reference?). Negative perceptions and widespread misinformation about dementia contribute to a lack of funding for essential services and support systems (Reference?). This underinvestment exacerbates existing inequalities, leaving people with dementia and their carers without the necessary resources to manage the condition effectively (Reference?)”

Response 4: Thank you for the comment. In the rewording of the text this element was integrated into the text differently.

Please, see p. 1, lines 37 – 39: “Public awareness and understanding of dementia remain frequently limited or distorted, contributing to widespread stigmatization and representing a significant barrier to accessing appropriate care (World Health Organization, 2021).”

Please, see p 2, lines 43 -46: “Stigma is acknowledged as one of the fundamental causes of health disparities influencing public policy and resource allocation, often resulting in the chronic underfunding of services dedicated to dementia care (Hatzenbuehler et al., 2013; Bayer, 2008).”

Comment 5: Similarly, some of the source article/s that illustrate the negative impact of stigma on dementia care (for PWD and Caregivers) are missing, and/or limited to the ADI report. You could, for example, briefly compare findings from ADI report with other findings. Also, see Brookman et al., 2025 for a detailed background on the literature relating to dementia-related stigma. 

Response 5: Thank you for the comment. Changes were made to the rewording of the text to incorporate more recent and more appropriate bibliographic references. With respect to the suggested reference, it was valuable in guiding the reasoning.

Comment 6: Page 3, paragraph 3, it is stated that, “To counteract this cycle, it is crucial to address both public and structural stigma.” However, structural stigma has not been defined. Please delete or include this earlier in the introduction when defining stigma.

Response 6: Thank you for your comment. In order not to generate further confusion, the term “structural stigma” was preferred to be deleted.

Comment 7: Overall, the introduction contains relevant content, but it is very lengthy, and repetitive. In addition, some of the content circles back on itself e.g., the relationship between dementia-related (DR) stigma and delayed diagnosis is mentioned multiple times in different ways and places. Revise to eliminate repetitions and consolidate ideas to improve conciseness, grammar and flow. For example, consider having a broad structure that takes the reader on a journey from a broad topic area (DR-stigma), to the narrower problem of the research gap (impact of DR-stigma on help seeking.; the global impact of dementia, definitions, public and self-stigma, the impact of dementia-related stigma on help seeking, and finally, current research aims.  

Response 7: Thank you for your comment. The introduction has been reorganized in order to improve flow and readability. In particular, valuable input from the following was taken into account in reorganizing the content.

Comment 8: Relatedly, there are some awkward transitions between ideas. Try and keep paragraphs and topic areas clearly defined, with transition sentences to the next idea. If the journal’s formatting guidelines permit it, subheadings can be helpful (written or imaginary).

Response 8: Thank you for your comment. In order to ensure greater connection between paragraphs and between proposed themes, the introduction has been reorganized following the suggested directions.

Minor points 

Comment 9: The introduction needs a thorough edit and checks for grammatical errors (e.g., "dependent by" should be "dependent on,") and typographical error (e.g., “an individuals”)

Response 9: Thank you for your comment. A check has been performed in order to resolve these issues.

Comment 10: ADI needs to be first spelt out first (i.e. Alzheimer’s Disease International [ADI], 2024) prior to using their acronym. 

Response 10: Thank you for your comment.

Methods 

Comment 11: Thematic synthesis is a good choice of analysis for qualitative studies. However, further detail is required e.g., was a deductive or inductive approach taken? Also, was open coding and software (e.g., NVIVO) used?

Response 11: Thank you for your comment. As suggested, more details have been provided with respect to the approach used and the possible use of software has been specified.

Please, see p. 5, lines 191 – 199: “The synthesis process followed three stages: (1) line-by-line coding of study findings, where each line of text was inductively coded by assigning descriptive labels to units of meaning (e.g., “Understanding of dementia”, “Impact on the person living with dementia”); (2) development of descriptive themes, in which similar codes were grouped together to reflect patterns across studies (e.g., “Lack of awareness”, “Perceived stigma”); (3) generation of analytical themes, where interpretations extended beyond the primary data to address the review question and offer new conceptual insights (e.g., “Reluctance to Share One’s Condition and Risk of Labelling”, “From Public Perception to Internalization or Rejection of Stigmatizing Beliefs”). No software was used during the process.”

Comment 12: For transparency, why did the final review only include qualitative studies? As this is mentioned later in the Discussion section as a limitation, it would seem relevant to mention why the decision was made to only include qualitative studies. 

Response 12: Thank you for the comment. For greater transparency, it was specified why only qualitative studies were included.

Please, see p. 4, lines 154 – 156: “Following the screening process, no eligible quantitative studies were identified; therefore, only qualitative studies meeting the inclusion criteria were incorporated into the final synthesis.”

Minor points 

Comment 13: The content in the Method section is appropriate and well structured. A thorough grammar check would improve readibility.

Response 13: Thank you for the comment. This was done in order to improve readability. 

Results  

Comment 14: The results section is comprehensive and with key components and themes and subthemes are well organised, and supported with relevant study citations. However, as with previous sections, revisions could be made to improve conciseness and grammar e.g., there is some switching between tense when reporting the findings.

Response 14: Thank you for the comment. This was done in order to improve conciseness and grammar.

Comment 15: A minor formatting point: the key table in the Results sections covers multiple pages (pages 9-15). If the width of the initial columns were narrowed, and the final column Headed “Key results” was expanded, the table would be shortened and easier to read. 

Response 15: Thank you for the comment. This was done in order to reduce the space occupied by Table 3.

Discussion 

Comment 16: The discussion section is well written and relates well to the existing literature. There were occasional exceptions to this that require minor amendments to improve accuracy e.g. Page 20, paragraph 2, “Literature indicates that stigmatizing attitudes are more pronounced among individuals with limited knowledge of AD, those with little contact with people with dementia, men, younger individuals, and those influenced by cultural interpretations of dementia (Hermann et al., 2018).” This statement is not entirely accurate and needs amending, as the literature indicates that the relationship between public DR-stigma and age is inconsistent. For example, there is a systematic review that has arrived at a different conclusion e.g., that DR-stigma is less pronounced in younger cohorts (see Nguyen and Li, 2020). Also, Brookman et al., 2025 found that dementia knowledge was only related to increased stigma in younger (not older) adults.

Response 16: Thank you for the comment. In order to reduce ambiguity, the quote has been removed and has been replaced by more relevant references.

Comment 17: As mentioned in the Method section, it would seem relevant to mention why the decision was made to only include qualitative studies in the Method section, to then mention this as a limitation. 

Response 17: Thank you for the comment. As a result of the change in the method section, the section on search limits has also been consistently changed.

Please, see p. 18, lines 616 - 620: “However, certain limitations warrant consideration when interpreting the findings. The inclusion of solely qualitative studies, while offering rich experiential data, constrains the generalizability of results and precludes quantitative assessment of stigma's impact on help-seeking behaviors.”

Round 2

Reviewer 2 Report

Comments and Suggestions for Authors

Than you for the opportunity to review revisions. The authors have adjusted the manuscript substantially to address all comments, suggestions, and concerns. 

Author Response

Dear Reviewer,
We sincerely thank you for your constructive feedback and appreciation of the revised manuscript.